# Type VI secretion system MIX-effectors carry both antibacterial and anti-eukaryotic activities

Ann Ray[1,2], Nika Schwartz[3], Marcela de Souza Santos[1], Junmei Zhang[1,2], Kim Orth[1,2,4] (iD) & Dor Salomon[3,*] (iD)

## Abstract

**Most type VI secretion systems (T6SSs) described to date are protein delivery apparatuses that mediate bactericidal activities. Several T6SSs were also reported to mediate virulence activities, although only few anti-eukaryotic effectors have been described. Here, we identify three T6SSs in the marine bacterium *Vibrio proteolyticus* and show that T6SS1 mediates bactericidal activities under warm marine-like conditions. Using comparative proteomics, we find nine potential T6SS1 effectors, five of which belong to the polymorphic MIX-effector class. Remarkably, in addition to six predicted bactericidal effectors, the T6SS1 secretome includes three putative anti-eukaryotic effectors. One of these is a MIX-effector containing a cytotoxic necrotizing factor 1 domain. We demonstrate that T6SS1 can use this MIX-effector to target phagocytic cells, resulting in morphological changes and actin cytoskeleton rearrangements. In conclusion, the *V. proteolyticus* T6SS1, a system homologous to one found in pathogenic vibrios, uses a suite of polymorphic effectors that target both bacteria and eukaryotic neighbors.**

**Keywords** bacterial competition; CNF1; *Vibrio*; virulence
**Subject Categories** Microbiology, Virology & Host Pathogen Interaction

## Introduction

A common strategy used by bacteria to manipulate neighbors is to secrete proteins, termed toxins or effectors, via specialized protein secretion systems [1]. Whereas some secretion systems secrete toxins to the extracellular space, others can deliver them directly into neighboring cells [1]. The type VI secretion system (T6SS) is a contact-dependent protein delivery apparatus found in many Gram-negative bacteria [2,3]. This system is composed of a tail-tube made of stacked hexameric rings of a protein called Hcp [3], which are capped by a spike complex made of a VgrG trimer [4] and a PAAR repeat-containing protein [5,6]. The tail-tube is decorated with

effector proteins that mediate the activity of the T6SS [7]. Upon activation, a sheath-like structure engulfing the tail-tube contracts and propels the tail-tube, with the effectors that decorate it, out of the bacterial cell and into an adjacent recipient, where effectors are deployed [6,8]. T6SSs are unique as they can directly deliver effectors into either bacterial or eukaryotic cells [9,10] and thus play a role in interbacterial competition or virulence, respectively [7].

T6SS effectors come in two flavors: as domains fused to tail-tube components, or as proteins that bind to these tail-tube components either directly [11,12] or via adapter proteins [13–16]. As many T6SSs described to date mediate antibacterial activities, most effectors that have been characterized are bactericidal and carry various activities such as lipases [17,18], nucleases [19], peptidoglycan hydrolases [20], pore-forming [21], and $NAD(P)^+$ glycohydrolases [22]. To prevent self-intoxication, these bactericidal effectors are encoded adjacent to cognate immunity genes [20,23]. Several T6SSs were shown to target eukaryotic cells, yet compared to the number of bactericidal effectors, few anti-eukaryotic effectors have been described to date [24]. These include toxin domains fused to "evolved VgrGs" [4,25], pore-forming VasX [21,26], PldB phospholipase [27], TecA deamidase [28], EvpP [29], and effectors secreted by the *Francisella* T6SS [30].

We recently uncovered a widespread, polymorphic class of T6SS effectors that share an N-terminal domain named MIX (*M*arker for type s*IX* effectors) and variable C-terminal toxin domains [31]. MIX-effectors group into five clans named MIX I–V [31]. Members of MIX V clan are horizontally shared between marine bacteria and are suggested to enhance their fitness [32]. In a recent report, we determined the secretome of *Vibrio proteolyticus* (hereafter referred to as *Vpr*), a marine bacterium that has been previously isolated from diseased corals [33,34]. We identified a functional hemolysin that is cytotoxic toward mammalian cells [35]. The *Vpr* secretome also included tail-tube components of T6SSs and MIX-effectors, indicating the presence of functional T6SSs.

Here, we set out to characterize the T6SSs of *Vpr* and determine their effector repertoires and activities. We found that of the three T6SSs encoded by *Vpr*, T6SS1 mediated antibacterial activities. Using comparative proteomics analysis to identify the T6SS1 effector repertoire, combined with functional assays, we found that in

1 Department of Molecular Biology, University of Texas Southwestern Medical Center, Dallas, TX, USA
2 Howard Hughes Medical Institute, University of Texas Southwestern Medical Center, Dallas, TX, USA
3 Department of Clinical Microbiology and Immunology, Sackler Faculty of Medicine, Tel Aviv University, Tel Aviv, Israel
4 Department of Biochemistry, University of Texas Southwestern Medical Center, Dallas, TX, USA
*Corresponding author. Tel: +972 3 6408583; E-mail: dorsalomon@mail.tau.ac.il

addition to antibacterial MIX- and non-MIX-effectors, T6SS1 also secretes MIX-effectors with anti-eukaryotic activities. Thus, *Vpr* employs T6SS1 and a suite of MIX-effectors to manipulate both bacterial and eukaryotic neighbors.

# Results and Discussion

## *Vpr* encodes three T6SSs and seven MIX-effectors

Genomic analysis of the *Vpr* strain ATCC 15338 (also called NBRC 13287) revealed three T6SSs encoded on three main gene clusters, each including the conserved core T6SS components and accessory genes [36] (Fig 1). The T6SS1 cluster (*vpr01s_06_00050–vpr01s_6_00300*) is homologous to the antibacterial *Vibrio parahaemolyticus* T6SS1 and *Vibrio alginolyticus* T6SS1, which we previously characterized [31,32]. It contains two putative transcription regulators (*vpr01s_06_00080* and *vpr01s_06_00230*) [37] and a MIX-effector/immunity (E/I) pair (*vpr01s_06_00050/vpr01s_06_00060*) [31]. However, it is missing the gene encoding a PAAR repeat-containing protein at the end of the cluster. The T6SS2 cluster (*vpr01s_08_01740–vpr01s_08_01940*) is similar to the T6SS3 cluster (*vpr01s_01_04280–vpr01s_01_05020*) in terms of gene content and organization, with two differences: (i) the genes encoding VgrG2-PAAR2 (*vpr01s_08_01940–vpr01s_08_01920*) are transcribed in the same direction as the rest of the cluster, whereas their counterparts in T6SS3 (*vpr01s_01_05000–vpr01s_01_05020*) are inverted; (ii) the genes between *clpV* and *fha* differ between the clusters (*vpr01s_08_01820–vpr01s_08_01800* in T6SS2; *vpr01s_01_04890–vpr01s_01_04880* in T6SS3) (Fig 1). Neither the T6SS2 nor the T6SS3 clusters appear to encode effectors.

The *Vpr* genome also contains "orphan" T6SS modules and effectors found outside the main T6SS clusters. An "orphan" *hcp4* (*vpr01s_11_00460*) encodes a protein 99% identical to Hcp1. Five "orphan" modules encode PAAR repeat-containing proteins which are either found upstream of putative antibacterial E/I pairs with predicted nuclease or lysozyme-like activities (*vpr01s_23_00260, vpr01s_15_00320,* and *vpr01s_04_02460*) or are fused to a putative C-terminal effector domain (*vpr01s_01_01270* and *vpr01s_06_01360*). There are also six "orphan" MIX-effectors, in addition to the MIX-effector found in the T6SS1 cluster. *vpr01s_06_01360* is a MIX II clan member that is fused to an N-terminal PAAR domain mentioned above. *vpr01s_28_00230* contains two MIX domains belonging to clans IV and V. *vpr01s_06_01710* and *vpr01s_25_00650* belong to the MIX V clan and carry putative antibacterial pore-forming (amino acid 502–584 similar to TcdA/TcdB pore-forming toxin domain with 46% probability, according to HHpred [38]) and Pyocin_S nuclease (according to NCBI conserved domains) toxin domains, respectively. Interestingly, two "orphan" MIX-effectors belonging to the MIX V clan appear to be anti-eukaryotic rather than antibacterial. *vpr01s_11_01570* contains a virulence CNF1 (cytotoxic necrotizing factor 1) domain, whereas *vpr01s_11_01580* does not contain a known C-terminal domain but is not part of an E/I pair as it has no adjacent immunity gene. An additional putative antibacterial effector with a DUF2235 lipase domain [18] is encoded by *vpr01s_10_00320* followed by three homologous putative immunity genes (*vpr01s_10_00310–vpr01s_10_00290*) (Fig 1). Interestingly, we found no "orphan" *vgrG* genes.

## *Vpr* kills competing bacteria under warm marine-like conditions

To characterize the *Vpr* T6SSs, we first set out to identify conditions that activate them. As our analysis revealed several T6SS effectors with predicted bactericidal activities, we reasoned that at least one of the three *Vpr* T6SSs will mediate antibacterial toxicity, which we can monitor as an indicator of T6SS activity. Therefore, we performed bacterial competition assays and monitored the viability of *Escherichia coli* prey (which is commonly used as a Gram-negative prey in bacterial competition assays) when incubated together with *Vpr* under different temperature (23, 30, or 37°C) and salinity conditions (1% or 3% NaCl). We found that *Vpr* bactericidal activity against *E. coli* prey was most prominent at 30°C on media containing 3% NaCl, as under these conditions we observed the largest decrease in *E. coli* prey viability (Fig 2). Thus, *Vpr* kills competing bacteria preferentially under conditions that are found in marine environments during summer months.

## *Vpr* bactericidal activity is mediated by T6SS1

To determine whether the bactericidal activity of *Vpr* is mediated by either of the T6SSs, we generated strains deleted for each of the three *vgrG* genes that correspond to the three T6SS clusters. We then tested the ability of these mutants to kill *E. coli* prey. Inactivation of T6SS1 (Δ*vgrG1*) resulted in complete loss of bactericidal activity, whereas inactivation of T6SS2 (Δ*vgrG2*) or T6SS3 (Δ*vgrG3*) had no effect (Fig 3A). These results suggested that T6SS1 was responsible for the *Vpr*-mediated bactericidal activity seen in Fig 2. Indeed, the expression and secretion pattern of the secreted T6SS1 tail-tube spike component VgrG1 agreed with the level of *Vpr* bactericidal activity under the same conditions and peaked at 30°C in media containing 3% NaCl (Appendix Fig S1). Attempts to rescue the bactericidal activity of Δ*vgrG1* by exogenous expression of VgrG1 from a plasmid were unsuccessful, possibly due to a polar effect of *vgrG1* deletion on downstream T6SS1 components (Appendix Fig S2). Therefore, we generated another T6SS1 mutant in which we deleted the gene encoding the essential T6SS baseplate component TssG1 (Δ*tssG1*) [39]. As with Δ*vgrG1*, the Δ*tssG1* mutant lost bactericidal activity against *E. coli* (Fig 3B) as well as against *V. parahaemolyticus*, a marine bacterium that is found in the natural habitat of *Vpr* (Fig 3C). Exogenous expression of TssG1 from a plasmid rescued the bactericidal activity and the secretion of the spike component VgrG1 (Fig 3B–D). Notably, deletion of *tssG1* had no adverse effects on *Vpr* growth (Appendix Fig S3). Thus, T6SS1 mediates the bactericidal activity of *Vpr* under warm marine-like conditions.

## The *Vpr* T6SS1 secretome includes multiple tail-tube components and putative antibacterial and anti-eukaryotic effectors

Next, we set out to determine the *Vpr* T6SS1 effectors that mediate the bactericidal activity. To this end, we used mass spectrometry to analyze and compare the secretomes of *Vpr* wild-type (T6SS1[+]) and Δ*tssG1* (T6SS1[−]) grown under T6SS1-inducing conditions (i.e., 30°C in media containing 3% NaCl). We identified 27 proteins that were predominantly found in the secretome of the wild-type *Vpr* (ratio of wild-type/Δ*tssG1* > 5) (Table 1 and Dataset EV1). Of these 27 proteins, 15 are predicted to be T6SS1 components and effectors (Fig 4), as they (i) are tail-tube components; (ii) contain MIX

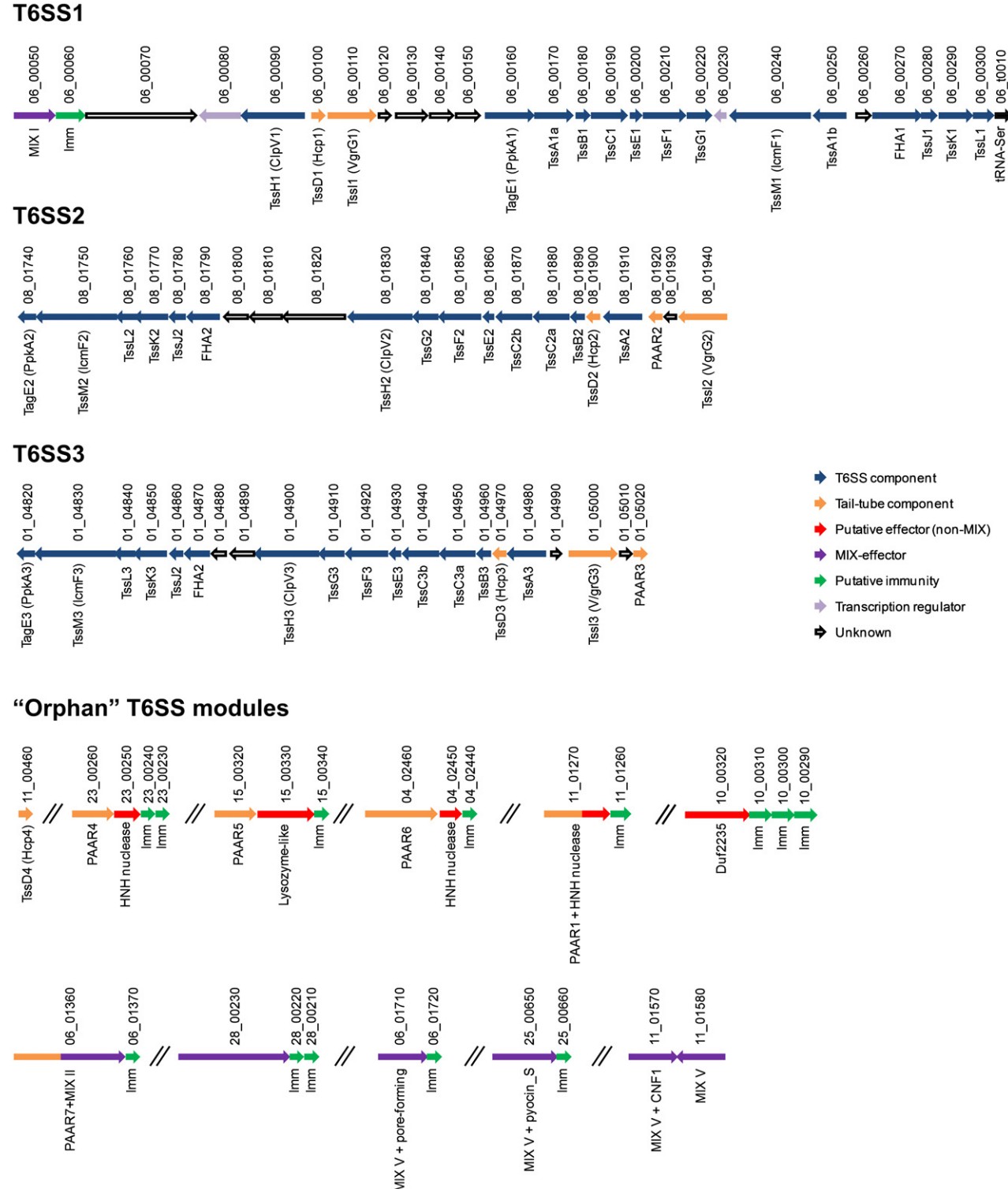

**Figure 1.  T6SS clusters and modules identified *in silico* in *Vpr*.**
Genes are represented by arrows indicating direction of translation. Locus tags (*uprO1s_xx_xxxxx*) shown above. Encoded proteins and known domains shown below.

domains; (iii) are homologs of known T6SS effectors; (iv) contain predicted toxin domains; (v) do not contain a canonical secretion signal; (vi) are in an operon with T6SS tail-tube components. As

expected, the tail-tube components Hcp1 and VgrG1, encoded within the T6SS1 gene cluster, were secreted in a T6SS1-dependent manner. Interestingly, the "orphan" Hcp4 and two "orphan" PAAR

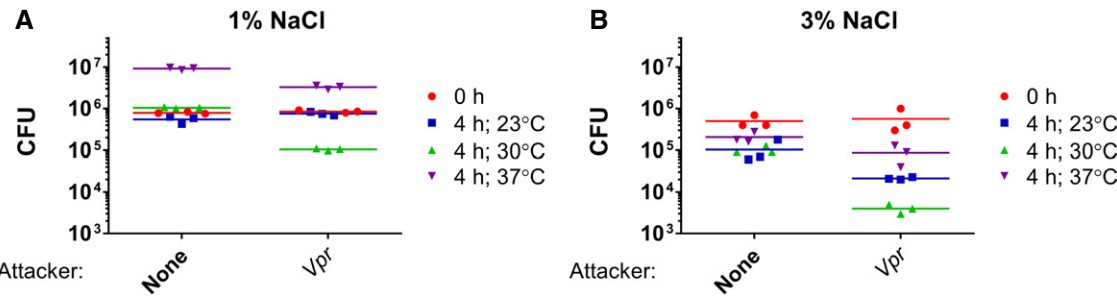

**Figure 2.  *Vpr* is bactericidal under warm marine-like conditions.**

A, B   Viability counts of *Escherichia coli* prey before (0 h) and after (4 h) co-culture with wild-type *Vpr* or with no attacker (none) at indicated temperatures on media containing 1% (A) or 3% (B) NaCl.

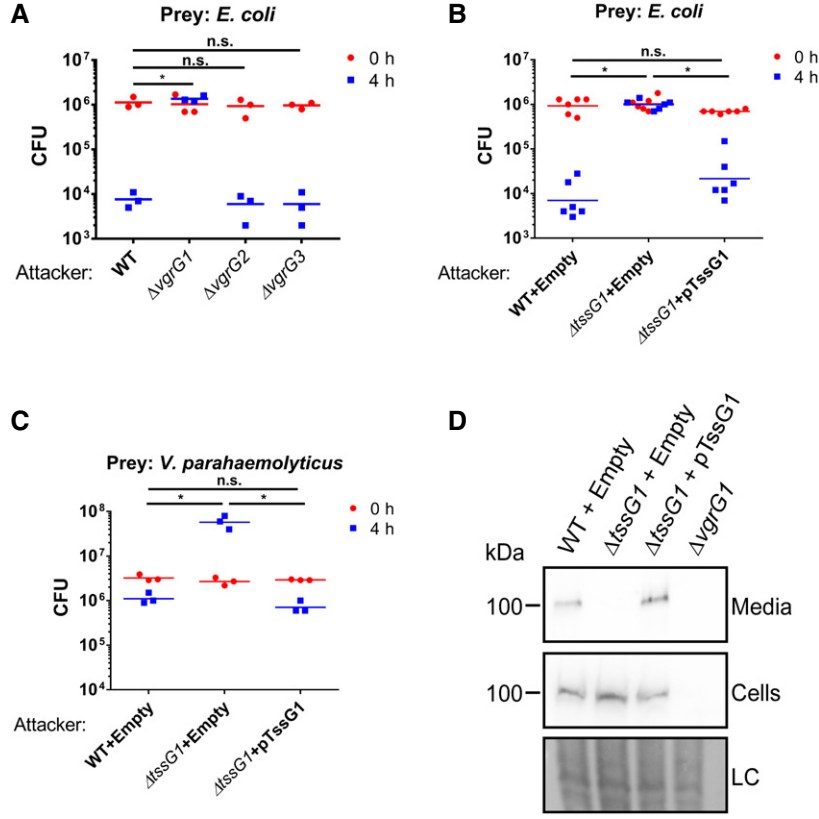

**Figure 3.  T6SS1 mediates *Vpr* bactericidal activity.**

A–C   Viability counts of *Escherichia coli* (A, B) and *Vibrio parahaemolyticus* POR1/Δ*hcp1* (C) prey before (0 h) and after (4 h) co-culture with indicated *Vpr* strains on media containing 3% NaCl at 30°C. Asterisks mark statistical significance between samples at 4-h timepoint by unpaired, two-tailed Student's *t*-test (*$P < 0.05$); n.s., no significant difference; WT, wild-type.

D   Expression (Cells) and secretion (Media) of VgrG1 were detected by immunoblot using specific antibodies against VgrG1. Loading control (LC) is shown for total protein lysate.

Data information: In (B–D), attackers contain either an empty expression vector (Empty) or vector for arabinose-inducible expression of TssG1 (pTssG1).
Source data are available online for this figure.

proteins (PAAR4 and PAAR6) were also identified as T6SS1 components [5].

We also detected nine potential effectors in the T6SS1 secretome, including five MIX-effectors (Table 1 and Fig 4). Six are predicted antibacterial effectors with various toxin domains and downstream small open reading frames that likely encode cognate immunity proteins (Figs 1 and 4, and Appendix Fig S4). Vpr01s_10_00320 contains a DUF2235 lipase domain; Vpr01s_04_02450 is a HNH-family nuclease (99% probability according to HHpred [38] analysis) encoded downstream of PAAR6 (Fig 1); Vpr01s_08_01700

**Table 1.  T6SS1-dependent *Vpr* secretome.**

| Function/Role | Uniprot | Locus [vpr01s_] | Description | Domains | Length [aa] | Signal peptide[a] |
|---|---|---|---|---|---|---|
| T6SS | | | | | | |
| Tail-tube components | U3BN98 | 06_00100 | Hcp1 | Hcp | 172 | No |
| | U3BB23 | 06_00110 | VgrG1 | VgrG | 692 | No |
| | U3BK42 | 11_00460 | Hcp4 | Hcp | 172 | No |
| | U3BHI2 | 23_00260 | PAAR4 | DUF4150 (PAAR) | 503 | No |
| | U2ZG25 | 04_02460 | PAAR6 | DUF4150 (PAAR) | 942 | No |
| Antibacterial effectors | U2ZJC6 | 10_00320 | Lipase | DUF2235 | 750 | No |
| | U2ZYX6 | 04_02450 | Nuclease | HNH/ENDO VII | 256 | No |
| | U3A2L0 | 08_01700 | Pore-forming | TcdA/B pore-forming domain-like (TM) | 356 | No |
| | U3BK37 | 06_00050 | MIX-effector | MIX I | 555 | No |
| | U2ZN70 | 25_00650 | MIX-effector nuclease | MIX V, Pyocin_S | 758 | No |
| | U3BSB9 | 28_00230 | MIX-effector | MIX IV, LysM, MIX V (TM) | 1,402 | No |
| Anti-eukaryotic effectors | U3BEL6 | 11_01580 | MIX-effector | MIX V | 591 | No |
| | U3BNK6 | 11_01570 | MIX-effector deamidase | MIX V, CNF1 | 603 | No |
| | U3A3W8 | 11_00400 | Txp40 homolog | | 90 | No |
| Unknown | U2ZZY6 | 06_00070 | VP1390 homolog | OmpA_C-like | 1,539 | No |
| Non-T6SS | | | | | | |
| Flagella | U3B8Q8 | 03_01380 | Flagellin | Flagellin_N, Flagellin_C | 288 | No |
| | U2ZXS3 | 03_01390 | Flagellar hook-associated protein 2 | FliD_N, FliD_C | 445 | No |
| | U3B8S7 | 03_01530 | Flagellar hook-associated protein 1 | FlgK | 457 | No |
| | U3BHW0 | 03_01570 | Flagellar basal body rod | FlgG | 261 | No |
| | U2ZXU5 | 03_01590 | Flagellar hook | FlgE | 398 | No |
| | U3BHV5 | 03_01520 | Flagellar hook-associated protein | FlgL | 299 | No |
| | U2ZYS5 | 03_01510 | Flagellin and related hook-associated protein | FlgL | 347 | No |
| Siderophore/heme transport | U3BJL7 | 05_01130 | Putative ferric siderophore receptor | TonB-dependent receptor | 663 | Yes |
| | U3A4E3 | 17_00240 | Ferrioxamine B receptor | TonB-dependent receptor | 710 | Yes |
| | U2ZCG7 | 01_01640 | Putative heme/hemoglobin receptor | TonB-dependent receptor | 711 | Yes |
| Other | U3BAK7 | 05_01090 | Hemolysin-type calcium-binding region homolog | WD-like beta propeller repeat | 463 | Yes |
| | U3A3U4 | 11_00150 | Protease | Peptidase_M28 | 596 | Yes |

TM, transmembrane.
[a]As determined by SignalP 4.1.

contains a C-terminal transmembrane region in a domain similar to pore-forming toxins (37% probability according to HHpred [38] analysis) (Table 1 and Fig 4) and is encoded close to the T6SS2 cluster (Appendix Fig S4). Three antibacterial effectors belong to the MIX-effector class: Vpr01s_06_00050 is encoded within the T6SS1 cluster and is homologous to the antibacterial MIX-effectors VP1388 and Va01565 from *V. parahaemolyticus* [31] and *V. alginolyticus* [32], respectively; Vpr01s_25_00650 belongs to the horizontally shared MIX V clan [32] and contains a Pyocin_S nuclease domain; Vpr01s_28_00230 contains both MIX IV and MIX V domains, as well as a C-terminal region homologous to that of the *Vibrio cholerae* pore-forming effector VasX [21] (Table 1 and Fig 4).

Surprisingly, this bactericidal T6SS1 also secreted three effectors with putative anti-eukaryotic activities, which are not part of bicistronic units that encode potential immunity proteins (Figs 1 and 4). Vpr01s_11_01570 (named Vpr01570 hereafter) belongs to the horizontally shared MIX V clan [32] and contains a CNF1 deamidase domain that is predicted to target Rho GTPases [40,41] (Appendix Fig S5). Vpr01s_11_01580 is encoded next to Vpr01570 and contains a MIX V domain (Fig 1). Its C-terminus is homologous to proteins encoded by bacteria of the genera *Vibrio*, *Morganella*, *Pseudomonas*, and *Yersinia*. Some of these homologs are annotated as cytotoxic and others contain Rhs repeats, which were suggested to be T6SS1 effectors [42]. Vpr01s_11_00400 is a small protein homologous to the C-terminus of the insecticidal toxin Txp40 [43] (Appendix Figs S4 and S6).

Another T6SS1-secreted protein is Vpr01s_06_00070. It is encoded downstream of the MIX-E/I pair Vpr01s_06_00050/Vpr01s_06_00060

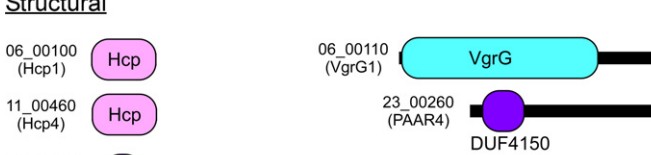

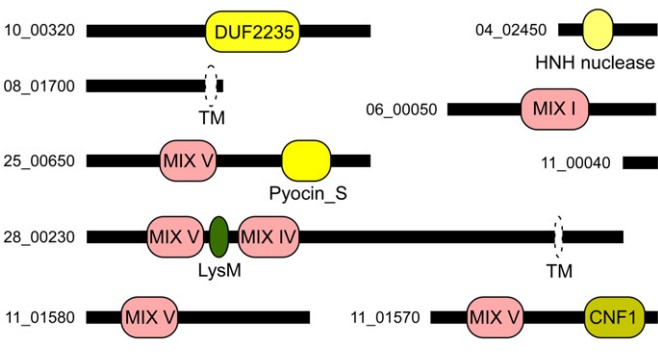

**Figure 4.   The *Vpr* T6SS1 secretome.**

Schematic representation of putative *Vpr* T6SS1-secreted proteins identified by comparative proteomics (see Table 1). Horizontal black bars are relative to protein length. MIX domains and domains identified in NCBI CDD are illustrated in color; domain names are inside or below the rectangles. Transmembrane helices (TM) identified in UniProt are illustrated as white ovals with dashed perimeters. DUF4150 belong to PAAR-like superfamily; DUF2235 is a phospholipase.

and is homologous to the *V. parahaemolyticus* VP1390, which we previously reported is secreted by the T6SS, yet its role remains unknown. Thus, the *Vpr* T6SS1 effector repertoire is diverse and includes multiple MIX-effectors with both antibacterial and anti-eukaryotic toxin domains. Notably, unlike the *Vpr* T6SS1, homologous systems in *V. parahaemolyticus* and *V. alginolyticus* secrete only one of each of the tail-tube components Hcp, VgrG, and PAAR [31,32]. Thus, it appears that the *Vpr* T6SS1 acquired additional "orphan" Hcp and PAAR proteins which possibly allow for diverse tail-tube combinations and enable secretion of a wide effector repertoire.

The remaining 12 proteins predominantly found in the wild-type *Vpr* secretome are likely not direct substrates of T6SS1. These include flagellar components or proteins that contain a signal peptide and are associated with siderophore/heme transport, a protein containing a WD-like beta propeller repeat, and a protease (Table 1). While we do not yet know what mechanism leads to this apparent down-regulation of the flagellar machinery and other non-T6SS proteins, others have previously reported similar findings. For example, the T6SS in *Citrobacter* was reported to modulate flagellar gene expression and secretion [44], as did the T6SS in *Ralstonia* [45]. Therefore, it is possible

that a common cross talk mechanism exists between T6SS activity and motility, and perhaps also between the T6SS and siderophore or heme transport.

**The T6SS1 MIX-effector Vpr01570 is a functional CNF1 toxin**

Our comparative proteomics analysis revealed that T6SS1, which mediates bactericidal activities (Fig 3), also secretes effectors with predicted anti-eukaryotic activities (Table 1). This led us to hypothesize that the *Vpr* T6SS1 can use its diverse effector repertoire to target both bacteria and eukaryotic neighbors. To determine whether T6SS1 can also manipulate eukaryotic cells, we focused our efforts on the T6SS1 MIX-effector Vpr01570 that contains a CNF1 toxin domain and set out to monitor its functionality. CNF1 toxins are deamidases that target and activate Rho GTPases [40,41]. As Rho GTPases are master regulators of the actin cytoskeleton, CNF1 toxins induce actin cytoskeleton rearrangements inside eukaryotic cells, which are readily visualized by microscopy [41].

We recently reported that *Vpr* induces cell death in human epithelial HeLa cells and in RAW 264.7 murine macrophages [35]. This cytotoxicity was mediated by a secreted pore-forming hemolysin named VPRH [35]. Strangely, however, unmasking this VPRH-mediated cytotoxicity by infecting HeLa cells with a Δ*vprh* mutant did not reveal any observable actin cytoskeleton rearrangements [35]. This result implied that the CNF1 MIX-effector is either not a functional toxin or is not delivered into HeLa cells during infection. We therefore tested whether Vpr01570 is a functional virulence toxin by exogenously expressing it in HeLa cells and in yeast. Expression of Vpr01570 fused to superfolder green fluorescent protein (sfGFP) induced dramatic actin cytoskeleton rearrangements in HeLa cells, including formation of actin stress fibers and ruffles which are hallmark phenotypes caused by CNF1 activity [40] (Fig EV1A). Vpr01570 was also toxic when expressed in yeast (Fig EV1B). These toxic effects were dependent on a functional CNF1 domain as they were not observed when a catalytic mutant (cysteine 450 to serine) (Appendix Fig S5) was expressed (Fig EV1A and B). Notably, expression of the Vpr01570 mutant form was detected in both HeLa cells and yeast, whereas the wild-type form was detected only in HeLa cells, presumably due to its deleterious effect in yeast (Fig EV1C and D). Moreover, we confirmed that deletion of *vprh* did not impair T6SS1 activity as a Δ*vprh* mutant remained bactericidal in a bacterial competition assay (Appendix Fig S7). Thus, Vpr01570 is a functional CNF1 toxin, yet *Vpr* does not use T6SS1 or Vpr01570 to manipulate the actin cytoskeleton during infection of HeLa cells.

**Vpr induces hemolysin-independent morphological changes in macrophages**

Previous reports suggested that the anti-eukaryotic activities of *V. cholerae* and *Burkholderia pseudomallei* T6SSs are activated upon internalization by phagocytic cells during infection [9,46]. As we did not observe T6SS1-mediated morphological changes during infection of HeLa cells with the Δ*vprh* mutant strain, we decided to test whether such effects will be detectable upon *Vpr* infection of phagocytic cells. To this end, we infected RAW 264.7 murine macrophages with *Vpr*. In agreement with our previous report [35], *Vpr* was cytotoxic to RAW 264.7 cells upon infection, and free nuclei from lysed cells were visible (Fig 5A). Remarkably, unmasking the

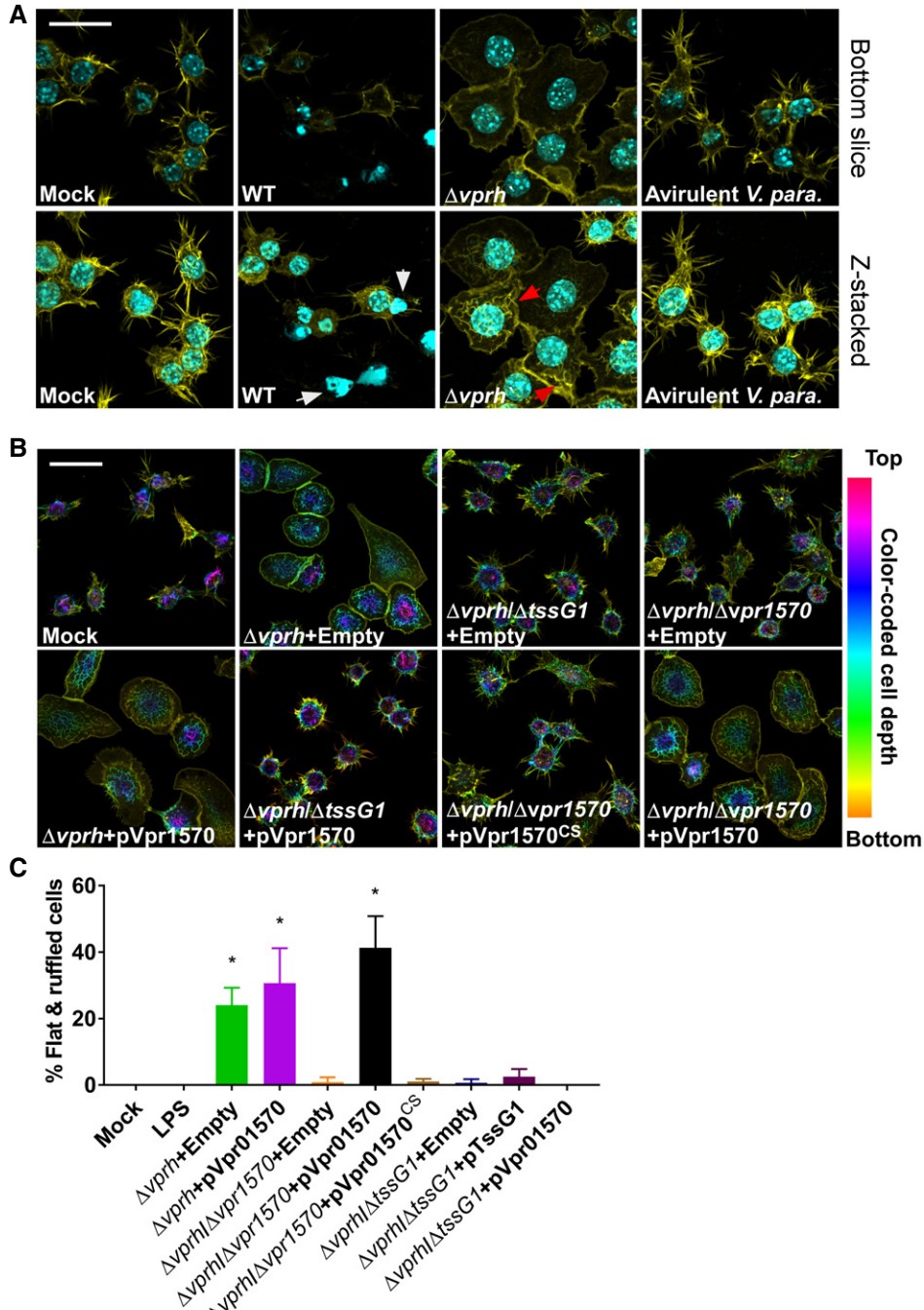

**Figure 5.  The T6SS1 MIX-effector Vpr01570 induces actin rearrangements in macrophages.**

A  Confocal micrograph of RAW 264.7 cells infected with indicated *Vpr* or avirulent *Vibrio parahaeolyticus* (*V. para*) strains for 4 h. Cells were stained for F-actin and DNA using rhodamine-phalloidin (yellow) and Hoechst stain (cyan), respectively. Top panels show bottom focal plane of the cells; lower panels show stacked *z*-axis slices of the same field of view. White arrows mark free nuclei of lysed cells. Red arrows mark actin ruffles. Scale bar = 20 μm; WT, wild-type.

B  Confocal micrograph of RAW 264.7 cells infected with indicated *Vpr* strains containing either an empty vector (Empty) or arabinose-inducible vectors for expression of wild-type Vpr1570 (pVpr1570) or C450S mutant form (pVpr1570$^{CS}$) for 4 h at MOI = 25. Cells were stained for F-actin using rhodamine-phalloidin. F-actin is color-coded to show cell depth in the *z*-axis. Scale bar = 20 μm; WT, wild-type.

C  Quantification of percentage of infected cells presenting a flatten and ruffled morphology after 6 h infection with the indicated strains or addition of LPS (0.1 μg/ml). Results shown as mean percentage of cells in a field ± standard deviation (*n* = 3, minimum number of cells per field is 40; maximum number of cells per field is 87). Asterisks mark treatments that are significantly different from LPS treatment as determined by unpaired, two-tailed Student's *t*-test (**P* < 0.05). Experiment was performed at least twice and a representative result is shown.

VPRH-mediated cytotoxicity by infecting macrophages with a Δ*vprh* mutant revealed dramatic morphological changes that were not detected upon infection of HeLa cells. RAW 264.7 cells infected with Δ*vprh* became flat, spread out, developed actin ruffles on the top side, and showed a pronounced and smooth actin border starting at 4 h post-infection (Figs 5 and EV2). Notably, infection of macrophages with an avirulent derivative of the bacterium *V. parahaemolyticus* [47] did not induce similar phenotypes (Fig 5A), nor did incubation with lipopolysaccharide (LPS) (Fig EV2). These results suggested that the phenotypes observed during infection with the *Vpr* Δ*vprh* mutant were induced by a *Vpr* determinant, and were not likely simply the result of macrophage activation by recognition of common pathogen-associated molecular patterns [48].

### *Vpr* uses the T6SS1 MIX-effector Vpr01570 to manipulate actin in macrophages

Next, we tested whether the *Vpr*-induced morphological changes in macrophages were mediated by T6SS1 activity. Inactivation of T6SS1 by deleting *tssG1* abolished the actin rearrangements seen upon infection of RAW 264.7 cells with the Δ*vprh* strain (Fig 5B and C). This result indicated that T6SS1 is required for the observed *Vpr*-induced phenotypes. Furthermore, the ability of *Vpr* to induce T6SS1-dependent actin rearrangements in macrophages but not in HeLa cells suggested that uptake of *Vpr* by a phagocytic cell may be a prerequisite for activation of T6SS1 during infection of eukaryotic cells. Surprisingly, exogenous expression of TssG1 from a plasmid did not restore the ability of the Δ*vprh*/Δ*tssG1* mutant to induce morphological changes in RAW 264.7 cells (Fig 5C). A possible explanation is that activation of T6SS1 during infection of macrophages requires precise regulation which is lost upon inducible overexpression of TssG1 from a plasmid.

Interestingly, actin reorganization phenotypes similar to those observed upon infection of RAW 264.7 cells with *Vpr* Δ*vprh* were previously reported upon treating macrophages with the *Yersinia* CNFy toxin [49], which is homologous to the CNF1 toxin domain of the T6SS1 MIX-effector Vpr01570 (Appendix Fig S5). Therefore, we asked whether Vpr01570 plays a role in the T6SS1-mediated actin phenotypes in macrophages. Indeed, deletion of *vpr01570* impaired the ability of *Vpr* Δ*vprh* to induce the T6SS1-mediated phenotypes in macrophages (Fig 5B and C). Exogenous expression of the wild-type Vpr01570, but not of its catalytic mutant form (Vpr01570[CS]), rescued the mutant strain's ability to induce actin rearrangements. However, overexpression of Vpr1570 did not induce similar phenotypes when T6SS1 was inactive (Fig 5B and C). Importantly, deletion of *vpr01570* did not impair T6SS1 functionality as the Δ*vprh*/Δ*vpr01570* mutant retained bactericidal activity, nor did it affect bacterial growth (Appendix Fig S8). These results demonstrate that T6SS1 delivers the MIX-effector Vpr01570 into macrophages where it employs its CNF1 toxin domain to induce morphological changes. Thus, *Vpr* can target both bacterial and eukaryotic neighbors with its T6SS1. This ability to use T6SS1 for delivery of polymorphic MIX-effectors into both bacteria and potential eukaryotic predator or host cells must provide a fitness advantage as it allows *Vpr* to manipulate neighbors which are present in its natural marine habitat [33,34].

Moreover, MIX-effector-secreting T6SSs homologous to the *Vpr* T6SS1 are present in many other marine bacteria, including emerging pathogens such as *V. parahaemolyticus* [37] and *V. alginolyticus*

[32]. We previously characterized MIX-effectors with antibacterial activities [31,32] and showed that effectors containing the MIX V domain are horizontally shared between marine bacteria and are used to diversify effector repertoires, thus enhancing competitive fitness [32]. Based on our current results, we further propose that bacteria can also acquire virulence MIX-effectors, which are evidently circulating in the *Vibrio* pan-genome, and use them to diversify their virulence traits and range.

After validating that Vpr01570 is an anti-eukaryotic MIX-effector delivered by T6SS1, we set out to determine whether this effector is not only necessary but also sufficient to induce the T6SS1-mediated morphological changes in macrophages. To this end, we transfected RAW 264.7 cells with an expression vector encoding Vpr01570 fused to sfGFP. Cells transfected with the wild-type Vpr01570 presented pronounced actin ruffling at the top of the cells which was not apparent upon expression of the catalytic mutant form Vpr01570[CS] (Fig EV3). Vpr01570 also co-localized with the actin ruffles (Fig EV3). However, Vpr01570-expressing cells did not spread or flatten as cells infected with *Vpr* Δ*vprh* did. These results suggested that whereas Vpr01570 is necessary and sufficient to induce actin ruffling in macrophages, it is not sufficient to cause cell spreading. Thus, it is possible that either additional effectors (of the T6SS1 or of other systems) or macrophage activation by recognition of pathogen-associated molecular patterns play a role, together with Vpr01570, in the induction of the observed morphological changes during infection. Notably, the *Vpr* T6SS3 may also be active under the same conditions as T6SS1. This is supported by the presence of the T6SS3 tail-tube proteins Hcp3 (Vpr01s_01_04970) and VgrG3 (Vpr01s_01_05000) in the wild-type *Vpr* secretome (Dataset EV1). As our results do not support a role for T6SS3 in bacterial competition, it is possible that this system targets eukaryotic cells. Further studies are required to determine the role and effector repertoire of T6SS3, as well as the mechanisms of action and targets of the T6SS1 putative anti-eukaryotic effectors Vpr01s_11_01580 and Vpr01s_11_00400.

In conclusion, this study demonstrates that *Vpr* uses its T6SS1 to mediate both bactericidal and virulence activities by delivering a suite of antibacterial and anti-eukaryotic effectors. The ability of the *Vpr* T6SS1 to deliver MIX-effectors into both bacteria and eukaryotic cells makes it, together with polymorphic members of the MIX-effector class, a lucrative candidate to use as a platform for biocontrol against pathogens from different kingdoms.

## Materials and Methods

### Bacterial strains, cell cultures, and yeast

*Vibrio proteolyticus* strain ATCC 15338 (also named NBRC 13287) (*Vpr*) and its derivative strains, as well as the *V. parahaemolyticus* RIMD 2210633 derivatives POR1/Δ*hcp1* (Δ*tdhAS*/Δ*hcp1*) [37] and POR4 (Δ*tdhAS*/Δ*vcrD1*/Δ*vcrD2*) [47], were routinely grown in marine Luria-Bertani (MLB) broth (Luria-Bertani broth supplemented with NaCl to a final concentration of 3 % w/v) at 30°C. To induce expression of genes from a plasmid, 0.1 % (w/v) L-arabinose was included in the medium. *Escherichia coli* strain DH5α were used for plasmid maintenance and amplification, and as prey in competition assays (see below). *Escherichia coli* strain S17-1 (λ *pir*) were used for

maintenance of pDM4 plasmids and mating (see below). *Escherichia coli* were routinely grown in 2× YT broth (1.6% w/v tryptone, 1% w/v yeast extract, 0.5% w/v NaCl) at 37°C. When necessary, media were supplemented with 30 μg/ml (for *E. coli*) or 250 μg/ml (for *Vpr*) kanamycin, or 25 μg/ml chloramphenicol. RAW 264.7 murine macrophages (ATCC TIB-71) and human epithelial HeLa cells (ATCC CCL-2) were purchased from ATCC and cultured in high-glucose Dulbecco's modified Eagle's medium [DMEM (Gibco)] supplemented with 10% (v/v) fetal bovine serum, 1% (v/v) penicillin–streptomycin–glutamine, and 1% (v/v) sodium pyruvate (termed supplemented DMEM) and kept at 5% $CO_2$ at 37°C. Cells have not been authenticated following receipt from ATCC. *Saccharomyces cerevisiae* strain BY4741 (MATa *his3Δ0 leu2Δ0 met15Δ0 ura3Δ0*) were grown in synthetic dropout media, as previously described [50].

## Plasmid construction

For arabinose-inducible expression in bacteria, the coding sequences (CDS) of VgrG1 (accession number GAD66994) and TssG1 (accession number GAD67005) were amplified from *Vpr* genomic DNA and inserted into the multiple cloning site (MCS) of the pBAD2 vector (Kan^R pBAD/*Myc*-His in which the *NcoI* restriction site had been removed [37]) in-frame with the C-terminal *Myc*-6xHis tag (producing pVgrG1 and pTssG1, respectively). For arabinose-inducible expression in bacteria, the CDS of Vpr01570 (accession number GAD68163), including the stop codon, was amplified from *Vpr* genomic DNA and inserted into the MCS of pMBAD (Kan^R pBAD/*Myc*-His in which an N-terminal c-*Myc* tag was inserted in the *NcoI* and *XhoI* restriction sites [35]), in-frame with the N-terminal c-*Myc* tag (producing pVpr01570). To generate a mutant form of Vpr01570 with a mutation in cysteine 450 to serine (pVpr01570^CS), site-directed mutagenesis was performed on pVpr01570 using a *pfu*-turbo DNA polymerase (Thermo Fisher Scientific) and the primers 5′-GGAACATTGAGTGGCaGCACCATGGTCACGGG-3′ and 5′-CCCGTGACCATGGTGCtGCCACTCAATGTTCC-3′, followed by *DpnI* treatment. The mutation was confirmed by nucleic acid sequencing.

For expression of Vpr01570 and Vpr01570^CS sfGFP-fusions in mammalian cells, the CDS of the wild-type Vpr01570 and the C450S mutant form were amplified from pMBAD vectors (described above) and inserted into the MCS of the sfGFP-N1 vector (Addgene) in-frame with the C-terminal sfGFP.

For galactose-inducible expression of Vpr01570 and Vpr01570^CS in yeast, the CDS of the wild-type Vpr01570 and the C450S mutant form were amplified from pMBAD vectors (described above) and inserted into the MCS of the pGMU10 (Riken) expression vector in-frame with the C-terminal c-Myc tag. The CDS of the *V. parahaemolyticus* type III secretion system effector VopR (*vp1683*), encoding amino acids 90–325 with a mutation in a catalytic cysteine 223 to alanine (VopR^Δ90/CA), was inserted into the MCS of the pDGFP vector [51] in-frame with the C-terminal eGFP-*Myc* tag, and used as a negative control for the yeast toxicity assay (see below).

## Construction of deletion strains

For in-frame deletions of *tssG1*(accession number GAD67005) and *vpr01570* (accession number GAD68163), 1-kb sequences directly upstream and downstream of each gene were cloned into pDM4, a

Cm^rOriR6K suicide plasmid [52]. These pDM4 constructs were inserted into *V. proteolyticus* via conjugation by S17-1(λ *pir*) *E. coli*. Transconjugants were selected for on MLB agar containing chloramphenicol. The resulting transconjugants were plated onto MLB agar containing 15% (w/v) sucrose for counter-selection and loss of the *sacB*-containing pDM4. Deletions were confirmed by PCR. The generation of in-frame deletions of *vprh* (accession number GAD67085), *vgrG1* (accession number GAD66994), *vgrG2* (accession number GAD67611), and v*grG3* (accession number GAD65726) were described previously [35].

## Bacterial competition

Bacterial strains were grown overnight in MLB (*Vpr* and *V. parahaemolyticus*) or 2× YT (*E. coli*). Cultures were normalized to $OD_{600} = 0.5$, mixed at 4:1 ratio (attacker:prey) and spotted on LB or MLB plates, as previously described [37]. Colony forming units (CFU) of the prey spotted at $t = 0$ h were determined by plating 10-fold serial dilutions on selective media plates. Bacterial spots were harvested from assay plates after 4-h incubation and the CFU of the surviving prey cells were determined. Assays were repeated at least twice with similar results, and results from a representative experiment are shown.

## Protein secretion

Bacterial strains were grown overnight in MLB with appropriate antibiotics to maintain plasmids when necessary. 0.9 $OD_{600}$ units were washed and re-suspended in 20 ml borosilicate glass tubes containing 5 ml of LB or MLB media (to produce cultures of $OD_{600} = 0.18$) with appropriate antibiotics to maintain plasmids, and with 0.1% (w/v) L-arabinose if induction of expression from a plasmid was required. Cultures were incubated with agitation (215 rpm) for 5 h at 23, 30, or 37°C, as indicated. For expression fractions (Cells) 1.0 $OD_{600}$ units were collected and cell pellets were re-suspended in 100 μl of Tris-glycine SDS sample buffer ×2 (Novex, Life Sciences). Supernatants of 10 $OD_{600}$ units were filtered and precipitated with deoxycholate and trichloroacetic acid [53]. Precipitated proteins were pelleted and washed twice with cold acetone, prior to re-suspension in 20 μl of 10 mM Tris–HCl pH = 8.0, followed by addition of 20 μl of 2× protein sample buffer. Expression and secretion samples were resolved on TGX stain-free gels (Bio-Rad) in SDS–polyacrylamide gel electrophoresis (SDS–PAGE), transferred onto PVDF membranes, and immunoblotted with custom-made anti-VgrG1 antibodies (polyclonal antibodies were produced in house (Thermo Scientific Pierce Protein Biology) with rabbits for peptide KDMSTKVLNNRYRDIGQDE). Loading of total protein lysate was visualized by analysis of trihalo compounds fluorescence of the immunoblot membrane (for Fig 3D) or Coomassie staining (for Appendix Fig S1).

## Bacterial growth

Overnight-grown cultures of *Vpr* were normalized to $OD_{600}$ of 0.1 in 25 ml MLB in triplicates and grown at 30°C with agitation for 6 h. Growth was assessed by $OD_{600}$ measurements at indicated time points. Experiments were performed at least twice with similar results. Results from a representative experiment are shown.

## Tissue culture infection

RAW 264.7 cells were seeded at $3 \times 10^5$ cells/ml in 6-well plates containing UV-pretreated 22- by 22-mm glass coverslips. Overnight *Vibrio* cultures were normalized to an $OD_{600}$ of 0.1 in 5 ml MLB and grown for 2 h with agitation at 30°C. When necessary, 200 µg/ml kanamycin and 0.1% (w/v) L-arabinose were added to maintain plasmids and induce expression. $OD_{600}$ measurements of the cultures were taken and used to calculate the volume needed to make solutions of DMEM without antibiotics (termed infection medium) containing a multiplicity of infection (MOI) of 25 for each bacterial strain (MOI of 10 was used for the *V. parahaemolyticus* avirulent POR4 strain, corresponding to the same $OD_{600}$ as MOI of 25 for *Vpr* strains). The infection solutions included 200 µg/ml kanamycin and 0.1% (w/v) L-arabinose when necessary. RAW 264.7 cells were washed twice with 1 ml of unsupplemented DMEM, and then 2 ml infection medium with bacteria were added to each well. The plates were centrifuged at $200 \times g$ for 5 min to synchronize infection before incubation at 37°C with 5% $CO_2$ for the indicated duration. Following infection, cells were washed once with 1× PBS (pH 7.4) and fixed to the coverslips by treating with 1 ml of 3.2% paraformaldehyde solution at 4°C overnight. Experiments were performed at least twice with similar results, and results from a representative experiment are shown.

## Tissue culture transfection

For transfections, HeLa cells were seeded either at $1 \times 10^5$ cells/ml (for immunoblots) or $5 \times 10^4$ cells/ml (for fluorescent microscopy), and RAW 264.7 cell were seeded at $5 \times 10^5$ cells/ml in 2 ml supplemented DMEM into 6-well plates containing UV-pretreated 22- by 22-mm glass coverslips. Culture media was changed 2 h prior to transfection. Cells were treated with addition of 200 µl OptiMEM containing 3:1 ratio of Fugene HD reagent (Promega) to plasmid DNA and cultured at 37°C. For immunoblots, HeLa cells were collected 16 h post-transfection and centrifuged at 1,000 rpm for 5 min. Pelleted cells underwent two washing steps in 1× PBS followed by centrifugation, then resuspended in 100 µl 2× protein sample buffer. Samples were resolved on SDS–polyacrylamide gel electrophoresis (SDS–PAGE), transferred onto PVDF membranes, and immunoblotted with anti-GFP antibodies (Living Colors A.v. Monoclonal antibody JL-8, Clontech). Equal loading of total protein lysate was confirmed by analysis of representative bands using Ponceau S staining of the immunoblot membrane. For microscopy, 7 h post-transfection cells were washed once with 1× PBS (pH 7.4) and then treated with 1 ml of 3.2% paraformaldehyde solution for 15 min at room temperature to fix cells to coverslips. The cells were then treated accordingly to the following Microscopy protocol.

## Microscopy

Fixed cells were treated with 0.1% Triton X-100 for 10 min to permeabilize membranes. The cells were washed with 1 ml 1× PBS and were subsequently stained with 200 µl of a dye solution for 10 min, covered. The dye solution contained a 1:1,000 dilution of 1 mg/ml Hoechst 33342 (Molecular Probes) and a 1:100 dilution of 200 U/ml rhodamine-phalloidin (Molecular Probes) to stain nuclei and F-actin, respectively. Cells were treated three

times to a 5-min wash with 1 ml 1× PBS to reduce background staining. After the final wash, the coverslips were mounted onto glass slides using a drop of ProLong Gold antifade mounting solution (Molecular Probes), and sealed after sitting at room temperature for several hours. Slides were imaged using a Zeiss LSM 710 or LSM 800 confocal microscope, and images were converted using ImageJ (NIH).

## Quantification of *Vpr*-induced morphological changes

Confocal *z*-stack images of F-actin for infected RAW 264.7 cells were stacked into a color-coded image using ImageJ temporal-color-coded hyperstacks, to allow for simultaneous assessment of cell height and F-actin morphology. Cells that presented altered morphology and were flat, spread out, developed actin ruffles on the top side, and showed a pronounced and smooth actin border were identified, and their percentage out of the total number of cells visible in the field of view was calculated. For each treatment, three fields of view were processed per experiment. Three independent experiments were performed for each treatment with similar results. Results from a representative experiment are shown.

## Yeast toxicity

pGMU10 and pDGFP expression vectors carrying Vpr01570 and VopR forms were transformed into the *S. cerevisiae* strain BY4741 by the lithium acetate transformation method [54]. Transformed yeast were plated onto selective synthetic dropout media lacking uracil and supplemented with 2% glucose. Growth-inhibition assays were performed as previously described [50]. Briefly, yeast cultures were grown overnight at 30°C in liquid selective media containing 2% glucose, washed, and normalized to an $OD_{600}$ of 1.0. Each strain was 10-fold serially diluted and spotted (10 µl) onto repressing (2% glucose) or inducing (2% galactose and 1% raffinose) solid selective media. Yeast were then incubated at 30°C for 2 days. Validation of protein expression in yeast was performed for 2 OD units of cells as previously described [50]. Proteins were detected using anti-c-Myc monoclonal antibodies (Santa Cruz, sc-40).

## Sequence analysis

T6SS clusters and "orphan" modules were identified by performing BLAST [55] searches using T6SS sequences from *V. parahaemolyticus* RIMD 2210633 [37]. MIX-effectors sequences were identified previously [31]. Domains within T6SS effectors were determined using the NCBI conserved domains database [56] or predicted using HHpred [38] and BLAST [55]. Secretion signals were predicted using the SignalP 4.1 server [57]. For multiple sequence alignments, protein sequences were aligned using T-coffee [58]. Alignments were visualized using Boxshade.

## Mass spectrometry

For secretome analysis, overnight cultures of wild-type or Δ*tssG1 Vpr* were normalized to an $OD_{600}$ of 0.18 in 50 ml MLB media in triplicates and grown for 5 h at 30°C with agitation. Media were collected, and proteins were precipitated as previously described

using deoxycholate and trichloroacetic acid [53]. Protein samples were run 10 mm into the top of an SDS–PAGE gel, stained with Coomassie blue, and excised.

Overnight in-gel digestion with trypsin (Promega) was performed after reduction and alkylation with DTT and iodoacetamide (Sigma-Aldrich). The resulting samples were analyzed by tandem MS using an LTQ Orbitrap Fusion Lumos ETD Tribrid mass spectrometer (Thermo Fisher Scientific) coupled to an EASY nLC1200-Nano liquid chromatography system (Thermo Fisher Scientific). Peptides were loaded onto a 75–μm i.d., 50-cm long, column containing 2 μm C18 resin (Thermo Fisher Scientific) and eluted with a gradient: 0–5% B in 2 min, 5–30% in 65 min; 30–60% B in 10 min. Buffer A consisted of 2% (v/v) acetonitrile (ACN) and 0.1% formic acid in water. Buffer B consisted of 80% (v/v) ACN, 10% (v/v) trifluoroethanol, and 0.1% formic acid in water. The Lumos mass spectrometer acquired up to 20 high-energy, collision-induced dissociation fragment spectra for each full spectrum acquired.

Raw MS data files were converted to a peak list format and analyzed using the central proteomics facilities pipeline (CPFP), version 2.0.3 [59,60]. Peptide identification was performed using the X!Tandem [61] and open MS search algorithm (OMSSA) [62] search engines against the *V. proteolyticus* NBRC 13287 protein database from UniProt. Fragment and precursor tolerances of 20 ppm and 0.5 Da were specified, and three missed cleavages were allowed. Carbamidomethylation of Cys was set as a fixed modification, with oxidation of Met set as a variable modification. Label-free quantitation of proteins across samples was performed using SINQ normalized spectral index software [63]. Protein identifications were filtered to an estimated 1% protein false discovery rate (FDR) using the concatenated target-decoy method [64], and proteins needed to be present in all three wild-type samples to be considered. Additional requirements of two unique peptide sequences per protein and two spectral counts per protein per wild-type sample replicate were imposed, resulting in a final protein FDR < 1%. Two-tailed, unequal variance Student *t*-test was used to calculate *P*-values. Samples in which no peptides were identified were set at value 0 for the analysis.

### Statistical analysis

Unless otherwise stated, data are shown as mean ± standard deviation of the mean. For bacterial competition assays, data are presented as individual data points and their mean. Statistical significance (*P* < 0.05) was determined by two-tailed, unpaired Student's *t*-test.

### Data availability

The data supporting the findings of this study are available within the paper and its supplementary information. The raw mass spectrometry data files and the corresponding peak lists were uploaded to the MassIVE dataset repository (https://massive.ucsd.edu/ProteoSAFe/dataset.jsp?task = fa38fde1dc3e43e3abc675600e588838) and are available under MassIVE ID: MSV000080535.

Expanded View for this article is available online.

### Acknowledgements
We thank the University of Texas Southwestern Medical Center Proteomics core for assistance with MS analysis. This project has received funding from the European Research Council (ERC) under the European Union's Horizon 2020 research and innovation program (Grant Agreement No. 714224) (D.S.), and by the NIH grant R01-AI087808, the Welch Foundation grant I-1561, and Once Upon a Time... (K.O.); K.O. is a Burroughs Welcome Investigator in Pathogenesis of Infectious Disease, a Beckman Young Investigator, and a W. W. Caruth, Jr., Biomedical Scholar, and has an Earl A. Forsythe Chair in Biomedical Science. D.S. is an Alon Fellow. The funders had no role in study design, data collection and analysis, decision to publish, or preparation of the manuscript.

### Author contributions
DS, AR, and KO designed the experiments. AR, DS, NS, MSS, and JZ performed experiments. DS, AR, JZ, and KO analyzed the data. DS wrote the manuscript.

### Conflict of interest
The authors declare that they have no conflict of interest.

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
