## [Review Process File · EMBO Reports]

Manuscript EMBO-2017-44226

Type VI secretion system MIX-effectors carry both anti-bacterial and anti-eukaryotic activities

Ann Ray, Nika Schwartz, Marcela de Souza Santos, Junmei Zhang, Kim Orth, Dor Salomon

Corresponding author: Dor Salomon, Tel Aviv University

Review timeline:

Submission date:	16 March 2017
Editorial Decision:	13 April 2017
Revision received:	21 July 2017
Editorial Decision:	07 August 2017
Revision received:	14 August 2017
Accepted:	16 August 2017

Editor: Achim Breiling

Transaction Report:

1st Editorial Decision

13 April 2017

Thank you for the submission of your research manuscript to EMBO reports. We have now received reports from the three referees that were asked to evaluate your study, which can be found at the end of this email.

As you will see, all three referees highlight the potential interest of the findings. However, all three referees have raised a number of concerns and suggestions to improve the manuscript, or to strengthen the data and the conclusions drawn. As the reports are below, I will not detail them here, but I think it will be important to strengthen the data on the physiological role of Vpr1570 (point 2 by referee #1), to test if endocytosis of bacteria is indeed required for the observed phenotypes (last major point of referee #2), and to provide more detailed information and/or perform experiments regarding the potential effector proteins or cellular targets (comment by referee #3 regarding Supplementary Figure 9). Further, the statistical analysis and quantification of the data need to be significantly improved (see comments by referee #2 and #3).

Given the constructive referee comments, we would like to invite you to revise your manuscript with the understanding that all referee concerns must be addressed in the revised manuscript and in a point-by-point response. Acceptance of your manuscript will depend on a positive outcome of a second round of review. It is EMBO reports policy to allow a single round of revision only and acceptance or rejection of the manuscript will therefore depend on the completeness of your responses included in the next, final version of the manuscript.

REFeree REPORTS

Referee #1:

Main findings of the study:

This study by Ray et al shows that *Vibrio proteolyticus* employs its T6SS1 for antimicrobial and anti-eukaryotic activities. Using proteomics, they identified nine putative effectors and demonstrated that one effector carrying the CNF1 toxin domain can cause actin rearrangement in macrophage cells. Many of the putative effectors contain the MIX domain that the authors published in previous studies. The data are solid with proper controls in place, and the manuscript is well prepared. However, this study is mostly descriptive lacking new mechanistic insights toward understanding T6SS delivery or in-depth functional analysis of its substrates. There are two major issues.

1. In the discussion, line 263, the authors claim that few anti-eukaryotic effectors have been described. This statement is not true since anti-eukaryotic T6SS effectors have been described in a number of T6SS models including *Vibrio*, *Pseudomonas aeruginosa*, *Aeromonas hydrophila*, and *Burkholderia*. In fact, a number of other key primary references are missing. For example, line 45, the authors should reference Vettiger and Basler (2016, Cell) showing T6SS can deliver effectors directly into bacterial cytoplasm, and Line37, (Pukatzki 2006 PNAS) and (Mougous 2006 Science) are the first two T6SS papers. The following papers should also be cited, (Liang 2015 PNAS) for adaptor, and (Dong 2013 PNAS) for lipase, and (Shneider 2013 Nature) for PAAR. Line 56 (Ma 2009) this reference is not appropriate and should be replaced with (Pukatzki 2007 PNAS). A number of exciting discoveries have been made in the last 5 years in the T6SS field. But unfortunately, the authors cited review papers more than they probably should.
2. The physiological role of Vpr1570 is not clear. In the macrophage model, the vpr1570 mutant had a moderate effect on actin but whether this causes cytotoxicity or its physiological relevance remains elusive. As an orphan MIX effector, Vpr1570 secretion may involve binding to VgrG, PAAR, or Hcp. However, this wasn't explored in the study.

Minor issues:

- Line 119: preformed → performed
- Line 154: never the less → nevertheless
- Line 160: duf2235 → DUF2235
- Line 223: Borkholderia → Burkholderia

Figure 3C. Are VgrG1 antibodies specific to only VgrG1 (ie no cross-reactivity with the other VgrGs?)

Line 154: Can you comment on why PAAR4 and PAAR6 may have a secretion signal? Is the secretion signal required for T6SS-dependent secretion?

Referee #2:

Authors show that T6SS-1 of *V. proteolyticus* kills *E. coli* and identify 15 proteins potentially secreted by T6SS-1, six predicted to target bacteria and three predicted to target eukaryotic cells. Authors then identify one effector, which is targeted to macrophages and induces morphological changes and actin rearrangements. The bacterial killing assays are well controlled and performed.

Major points:

- Proteins secreted by T6SS were identified as those present in more than 5-times higher amount in the supernatant of T6SS+ than of T6SS- cells. An estimate of statistical significance for each protein is, however, missing and should be provided in the supplementary dataset (p-value based on biological replicates).
- Surprisingly, many proteins related to flagella as well as other proteins were differentially present in the T6SS+ and T6SS- samples. Authors should elaborate why this could be the case.

- An analysis of the morphological changes in eukaryotic cells caused by the identified effector is interesting, however, proper quantification is lacking. The observed phenotypes should be defined in more details and sufficient number of macrophages should be scored. A statistical analysis should be provided for the compared strains to conclude that the observed phenotypes are reproducible and robust. Authors should also attempt to block phagocytosis by any commercially available inhibitor and test if the observed phenotypes indeed require endocytosis of the bacteria.

Minor points:

The first paragraph of the introduction should reference more primary literature, not just reviews.

Referee #3:

This manuscript is based on overall solidly performed work (has a very high technical as well as scientific standard) and was well summarized in writing. The conclusions are largely backed up by the results. The work's rationale relies mainly on sequence homology to attribute potential functions to putative effectors in the genome of the marine bacterium Vpr and does not present a wealth of novel mechanisms. It is based on a comprehensive genomic analysis and summarizes T6SS in the Vpr species and in the genus *Vibrio* for numerous readers interested in these fields. It discovers that one of the T6SS(1) can transport antibacterial as well as eukaryotic effectors, so it targets two kingdoms at the same time. It will be very interesting to know which species it may target in the natural habitat. Maybe it can target coral cells directly, could be related to current increase in ocean temperature. It is certainly opening a new and relevant field of research. The manuscript is overall well written and concise.

General Comments:

The authors identify so called "marine conditions" as active conditions for the T6SS1. Could they highlight in the results' text which other conditions they tested for activity and why they state and conclude that only the MLB/marine condition activated the T6SS1? Expression data for T6SS1 structural proteins or cDNA under certain conditions are lacking, except for the proteins identified in the secretome analysis or overexpressed in cells (Western blot data). Authors should highlight the conditions under which the secretome analysis was performed in the results text (in addition to the methods).

In general, throughout the manuscript, morphological changes in target cells caused or not caused by the Vpr effects are not properly quantitated; the conclusions rely solely on one single image shown as photograph per condition. A scientifically correct way to describe these changes should be to use cellular parameters such as cell shape and diameter and quantitate them by exact measurements (of at least 50 or 100 cells picked in a non-biased manner under each condition).

Fig. 2 and results text: is *E. coli* occurring in the natural environment of Vpr? Why was *E. coli* chosen as a competitor? Is *E. coli* a proxy for other bacteria occurring in the natural environment? Does Vpr attack other *Vibrio* species? Please answer and explain rationale in results text.

Fig. 3 and results text: antibacterial effector causing the anti *E. coli* effect is not identified. Is it possible to at least speculate about the potential antibacterial effector causing the effect? Or is it likely that the effect on *E. coli* is a multifactorial effect?

Fig. 4 and text: it seems not to be excluded by the performed controls that T6SS1/vpr01570 activity is linked to macrophage activation by pattern recognition receptors or in conjunction with PRR cell activation. It might be advisable to do another control, such as siRNA of potential cellular targets of effector in macrophage. It is also possible by novel transfection routines (e.g. nucleofector) to transfect Raw macrophages quite successfully with plasmids. This approach might also help to pin down the vpr01570-related activity on the cells unambiguously.

Line 157: "identified nine putative effectors in the T6SS1 secretome" identified seems too strong a term (even if "putative" is used), since the described effects are not attributed directly to one of the potential effectors; as such, they remain functional orphans and are purely associated by homology.

So it is not identification but sequence-based provisional assignment. It is advisable to use another term, such as "detected potential effector proteins in the bacterial supernatants"

Supplementary Figure 8: the reviewer does not see a morphological change in the macrophage images of this figure and experimental series taken over time. Could the change be quantitated and statistically affirmed (e.g. measuring cell diameter, length or other changes of at least 50 or 100 cells and depicting this in a graphical rendering?)

Supplementary Fig. 9: Here, the phenotype change in the images, dependent on mutations of T6SS1 is very clear; however, some more detailed information and experiments, either with regard to the potential effector protein(s) causing these changes or the cellular target might just give the manuscript a much higher novelty aspect. Otherwise, the manuscript relies very much on sequence homology to proteins of other bacteria T6SS that have been described and characterized before.

Minor comments:

The manuscript contains a number of extensive lists that refer to the findings of genome analyses and secretome analyses, respectively. These lists render the manuscript quite difficult to read. In addition to the table (Table 2), it is recommended to present the putative effectors identified by the proteome analysis as a graphic scheme in the main text, highlighting the protein domains of all putative effectors of T6SS1 including the MIX class.

Line 223: „Borkholderia" should read Burkholderia

Effector classes are designated as "clans" throughout the manuscript. Might be better to use the term class?

Page 14: expression of effector in Hela cells and yeast cells: detecting expression of the effector is missing. Please add data in results text.

A number of text spaces are missing in front of reference brackets. Should be inserted.

1st Revision - authors' response

21 July 2017

Point-by-point response to reviewers' comments:

Referee #1:

Main findings of the study:

This study by Ray et al shows that *Vibrio proteolyticus* employs its T6SS1 for antimicrobial and anti-eukaryotic activities. Using proteomics, they identified nine putative effectors and demonstrated that one effector carrying the CNF1 toxin domain can cause actin rearrangement in macrophage cells. Many of the putative effectors contain the MIX domain that the authors published in previous studies. The data are solid with proper controls in place, and the manuscript is well prepared. However, this study is mostly descriptive lacking new mechanistic insights toward understanding T6SS delivery or in-depth functional analysis of its substrates. There are two major issues:

1. In the discussion, line 263, the authors claim that few anti-eukaryotic effectors have been described. This statement is not true since anti-eukaryotic T6SS effectors have been described in a number of T6SS models including *Vibrio*, *Pseudomonas aeruginosa*, *Aeromonas hydrophila*, and *Burkholderia*.

Indeed, several anti-eukaryotic T6SS effectors have been described, and we cite the appropriate work in the Introduction section (lines 55-59). Our intention is to emphasize that the number of virulence effectors which have been discovered is low compared to the number of bactericidal T6SS effectors. We amended the sentence in the Discussion to clarify this, and it now states: "Notably, compared with the high number of bactericidal T6SS effectors identified to date, only few T6SS effectors with anti-eukaryotic activities have been described" (lines 286-288), as well as

in the Introduction: “Several T6SSs were shown to target eukaryotic cells, yet compared to the number of bactericidal effectors, few anti-eukaryotic effectors have been described to date” (lines 55-57).

In fact, a number of other key primary references are missing. For example, line 45, the authors should reference Vettiger and Basler (2016, Cell) showing T6SS can deliver effectors directly into bacterial cytoplasm, and Line 37, (Pukatzki 2006 PNAS) and (Mougous 2006 Science) are the first two T6SS papers. The following papers should also be cited, (Liang 2015 PNAS) for adaptor, and (Dong 2013 PNAS) for lipase, and (Shneider 2013 Nature) for PAAR. Line 56 (Ma 2009) this reference is not appropriate and should be replaced with (Pukatzki 2007 PNAS). A number of exciting discoveries have been made in the last 5 years in the T6SS field. But unfortunately, the authors cited review papers more than they probably should.

We thank the reviewer for bringing this to our attention. We have amended the Introduction so that it now includes the suggested citations as well as additional ones that are relevant.

2. The physiological role of Vpr1570 is not clear. In the macrophage model, the vpr1570 mutant had a moderate effect on actin but whether this causes cytotoxicity or its physiological relevance remains elusive. As an orphan MIX effector, Vpr1570 secretion may involve binding to VgrG, PAAR, or Hcp. However, this wasn't explored in the study.

In previous work, we showed that in the absence of the pore-forming hemolysin VPRH Vpr is not cytotoxic toward RAW 264.7 macrophages (Ray et al., mBio, 2016), indicating that Vpr01570 does not induce cell death in this model. For the current study, we have tested whether Vpr01570 affects Vpr survival within macrophages and whether it mediates protection against predation by grazing amoeba, yet have yet to identify a significant effect in these systems (data not shown). The physiological role of Vpr01570 is thus currently under investigation in our lab.

As for the mechanism that governs secretion of Vpr01570 through the T6SS, we are actively pursuing this question as part of a larger project aimed at determining whether there is a unifying mechanism for secretion of MIX-effectors. Therefore, we think that attempting to solve the secretion mechanism of a single MIX-effector as part of the current study will not provide significant insight or contribute to the focus of the current work.

Minor issues

Line 119: preformed → performed

Line 154: never the less → nevertheless

Line 160: duf2235 → DUF2235

Line 223: Borkholderia → Burkholderia

These typos were amended.

Figure 3C. Are VgrG1 antibodies specific to only VgrG1 (ie no cross-reactivity with the other VgrGs?)

Yes, the antibody is specific to VgrG1 as it was raised against a short peptide not found in either VgrG2 or VgrG3 (for details, please refer to Methods section lines 435-437). The specificity can also be seen in Fig. 3D and Appendix Fig. S1 as the signal disappears in a strain deleted for vgrG1. We have attached a figure supporting the specificity of the antibody towards VgrG1 below.

Line 154: Can you comment on why PAAR4 and PAAR6 may have a secretion signal? Is the secretion signal required for T6SS-dependent secretion?

Upon closer inspection, we realized that these PAAR proteins were marked as positive for a secretion signal only due to similarity with other secreted proteins as determined by SignalBLAST. We now used SignalP to predict secretion signal as this is a commonly used prediction server. This new analysis in SignalP shows that PAAR4 and PAAR6 are not predicted to contain a secretion signal. The text has been amended (lines 162-163) as well as Table 1.

Referee #2:

Authors show that T6SS-1 of *V. proteolyticus* kills *E. coli* and identify 15 proteins potentially secreted by T6SS-1, six predicted to target bacteria and three predicted to target eukaryotic cells. Authors then identify one effector, which is targeted to macrophages and induces morphological changes and actin rearrangements. The bacterial killing assays are well controlled and performed.

Major points:

- Proteins secreted by T6SS were identified as those present in more than 5-times higher amount in the supernatant of T6SS+ than of T6SS- cells. An estimate of statistical significance for each protein is, however, missing and should be provided in the supplementary dataset (p-value based on biological replicates).

We now include statistical analysis in Dataset EV1.

- Surprisingly, many proteins related to flagella as well as other proteins were differentially present in the T6SS+ and T6SS- samples. Authors should elaborate why this could be the case.

Several other groups have previously reported an apparent cross-talk between T6SS activity and expression levels of the flagellar machinery in various bacteria (Zhang et al., Microb. Pathog., 2014; Liu et al., Infect. Immun., 2015). We have added a paragraph regarding this finding to the Discussion section (lines 314-321).

- An analysis of the morphological changes in eukaryotic cells caused by the identified effector is interesting, however, proper quantification is lacking. The observed phenotypes should be defined in more details and sufficient number of macrophages should be scored. A statistical analysis should be

provided for the compared strains to conclude that the observed phenotypes are reproducible and robust.

[Data not included in review process report.]

We added quantifications for percentage of cells presenting the described morphological changes. These can now be seen in Fig. 5C and Fig. EV2. Methods describing how cells were quantified were also added (Methods section, lines 492-500), as well as a more elaborate description of the observed phenotypes (lines 238-240).

Authors should also attempt to block phagocytosis by any commercially available inhibitor and test if the observed phenotypes indeed require endocytosis of the bacteria.

We have performed the experiment proposed by the reviewer. Indeed, we no longer observe the morphological changes mediated by Vpr1570 upon infection of macrophages when we inhibit phagocytosis with the common inhibitor cytochalasin D (see attached Figure below). However, we are facing an issue that hampers our ability to draw reliable conclusions from this experiment as cytochalasin D, similar to other inhibitors of phagocytosis, inhibits phagocytosis via inhibition of actin polymerization. Because the Vpr-mediated effect we observe during infection of macrophages is changes in the actin cytoskeleton which rely on the CNF1 activity of the effector Vpr01570 (that is predicted to manipulate Rho GTPases which are master regulators of the actin cytoskeleton), we cannot conclude what is the cause of the loss of morphological phenotypes. i.e. whether indeed inhibition of phagocytosis resulted in loss of T6SS1-mediated effects or whether we simply no longer observe the morphological changes because cytochalasin D inhibited actin polymerization. Therefore, we prefer to exclude this result from the current manuscript and continue to investigate the mechanism that governs the activation of the T6SS1-induced phenotype in RAW 264.7 cells for a later report.

Minor points:

The first paragraph of the introduction should reference more primary literature, not just reviews.

We thank the reviewer for bringing this to our attention. We have amended the Introduction so that it now includes the relevant citations.

Referee #3:

This manuscript is based on overall solidly performed work (has a very high technical as well as scientific standard) and was well summarized in writing. The conclusions are largely backed up by the results. The work's rationale relies mainly on sequence homology to attribute potential functions to putative effectors in the genome of the marine bacterium Vpr and does not present a wealth of novel mechanisms. It is based on a comprehensive genomic analysis and summarizes T6SS in the Vpr species and in the genus *Vibrio* for numerous readers interested in these fields. It discovers that one of the T6SS(1) can transport antibacterial as well as eukaryotic effectors, so it targets two kingdoms at the same time. It will be very interesting to know which species it may target in the natural habitat. Maybe it can target coral cells directly, could be related to current increase in ocean temperature. It is certainly opening a new and relevant field of research. The manuscript is overall well written and concise.

General Comments:

The authors identify so-called "marine conditions" as active conditions for the T6SS1. Could they highlight in the results' text which other conditions they tested for activity and why they state and conclude that only the MLB/marine condition activated the T6SS1?

The results text includes a description of the different temperature and salinity conditions used to test for Vpr antibacterial activity. We have also rephrased the Results paragraph to clarify the conditions and conclusion (lines 117-127).

Expression data for T6SS1 structural proteins or cDNA under certain conditions are lacking, except for the proteins identified in the secretome analysis or overexpressed in cells (Western blot data).

We now present expression and secretion data for the T6SS1 structural secreted protein VgrG1 under the different temperature and salinity conditions that we used to determine the Vpr bactericidal activity (Appendix Fig. S1, lines 135-138). We show that the pattern of VgrG1 secretion is in agreement with the Vpr bactericidal activity identified in Fig. 2.

Authors should highlight the conditions under which the secretome analysis was performed in the results text (in addition to the methods).

We added a description of the conditions used for the secretome analysis to the text (lines 153-155).

In general, throughout the manuscript, morphological changes in target cells caused or not caused by the Vpr effects are not properly quantitated; the conclusions rely solely on one single image shown as photograph per condition. A scientifically correct way to describe these changes should be to use cellular parameters such as cell shape and diameter and quantitate them by exact measurements (of at least 50 or 100 cells picked in a non-biased manner under each condition).

We thank the reviewer for this comment. Quantifications of cells showing the described morphology were added to Fig. 5 (Fig. 5C) and Fig. EV2 (Fig. EV2B). As parameters such as cell shape and diameter vary within each sample dependent on the local cell density, we have scored the cells visually based on actin morphology and cell shape. A “positive” macrophage showing the T6SS1-mediated phenotype was determined based on the relative height of the cell (compared to other cells in the field using Z-stack depth color coded actin), together with the appearance of a pronounced smooth actin border around the cell and ruffles on top. Flat cells that did not show smooth actin cell periphery and ruffles (which were sometimes seen in cultures treated with LPS or T6SS1 bacteria) were not counted as “positive” (detailed explanation can be found in the Results section lines 238-240, and Methods section lines 492-500).

Fig. 2 and results text: is *E. coli* occurring in the natural environment of Vpr? Why was *E. coli* chosen as a competitor? Is *E. coli* a proxy for other bacteria occurring in the natural environment? Does Vpr attack other *Vibrio* species? Please answer and explain rationale in results text.

We used E. coli as prey because it does not divide over the course of our experiment at 23 or 30°C, nor when grown on 3% NaCl (Salomon et al., PLoS One, 2013). Its inability to divide under most of our tested conditions allows us to clearly determine whether the attacking strain actually kills the prey (as we can clearly see reduction in prey viability that isn't always clearly apparent if the prey survivors continue to divide during the experiment.) We now include additional explanation in the text (lines 120-123), and we have also added a bacterial competition assay using Vibrio parahaemolyticus, another marine bacterium, as prey to show the Vpr bactericidal activity is relevant also against bacteria sharing its habitat (Fig. 3C).

Fig. 3 and results text: antibacterial effector causing the anti *E. coli* effect is not identified. Is it possible to at least speculate about the potential antibacterial effector causing the effect? Or is it likely that the effect on *E. coli* is a multifactorial effect?

Previous work showed that antibacterial activities of T6SSs are often mediated by several effectors, each carrying bactericidal activities (e.g. Hood et al., Cell Host Microbe, 2010; Fritsch et al., Mol Cell Proteomics, 2013; Dong et al., PNAS, 2013; Salomon et al., PNAS, 2014). Here, we detected six predicted T6SS1 effectors with putative antibacterial activities. Two of these effectors are homologs of bactericidal effectors of T6SS in V. parahaemolyticus that we previously described (Salomon et al., PNAS, 2014). Therefore, it is highly likely that several or all of these putative antibacterial effectors mediate bactericidal activities simultaneously during inter-bacterial competition.

Fig. 4 and text: it seems not to be excluded by the performed controls that T6SS1/vpr01570 activity is linked to macrophage activation by pattern recognition receptors or in conjunction with PRR cell

activation. It might be advisable to do another control, such as siRNA of potential cellular targets of effector in macrophage. It is also possible by novel transfection routines (e.g. nucleofector) to transfect Raw macrophages quite successfully with plasmids. This approach might also help to pin down the vpr01570-related activity on the cells unambiguously.

We thank the reviewer for this suggestion. To address these questions, we transfected RAW 264.7 cells with constructs expressing Vpr01570 (either wild-type or catalytic mutant) fused to a C-terminal sfGFP. We now show that Vpr01570 is sufficient to induce actin ruffling in RAW 264.7 cell, yet it does not seem to be sufficient to cause cell flattening and spreading (Fig. EV3). Thus, we cannot exclude a role for PAMPs of other effectors in inducing these phenotypes together with Vpr01570. We discuss these new results and their meaning in the text (Results section lines 267-281, and Discussion section lines 330-333).

Line 157: "identified nine putative effectors in the T6SS1 secretome" identified seems too strong a term (even if "putative" is used), since the described effects are not attributed directly to one of the potential effectors; as such, they remain functional orphans and are purely associated by homology. So it is not identification but sequence-based provisional assignment. It is advisable to use another term, such as "detected potential effector proteins in the bacterial supernatants"

Following the reviewer's suggestion, we changed "identified nine putative effectors" to "detected nine potential effectors" (line 164-165). We do want to emphasize that their assignment as effectors is not solely based on sequence homology, but is attributed also by our findings that they are secreted to the media in a T6SS1-dependent manner.

Supplementary Figure 8: the reviewer does not see a morphological change in the macrophage images of this figure and experimental series taken over time. Could the change be quantitated and statistically affirmed (e.g. measuring cell diameter, length or other changes of at least 50 or 100 cells and depicting this in a graphical rendering)?

We revised this figure (currently Fig. EV2) and it now includes quantification of the phenotypes. The morphological changes can be visualized more clearly with higher magnification as shown in Fig. 5 where the Vpr strain deleted for vprh was used.

Supplementary Fig. 9: Here, the phenotype change in the images, dependent on mutations of T6SS1 is very clear; however, some more detailed information and experiments, either with regard to the potential effector protein(s) causing these changes or the cellular target might just give the manuscript a much higher novelty aspect. Otherwise, the manuscript relies very much on sequence homology to proteins of other bacteria T6SS that have been described and characterized before.

Regarding the potential effector causing the changes shown in the (former) Supplementary Fig. 9 - we demonstrate in Fig. 5 that the morphological phenotypes that require a functional T6SS1 also require the T6SS1 MIX-effector Vpr01570. We further show that the T6SS1 effector Vpr01570 contains a CNF1 toxin domain known to target Rho GTPases, and that a functional CNF1 domain is necessary to cause these morphological phenotypes (as a point mutation in a catalytic cysteine of the CNF1 deamidase domain is unable to rescue the deletion strain). Based on these results, we proposed that T6SS1 delivers Vpr01570 to target Rho GTPases via its CNF1 toxin domain, and this in turn results in the morphological changes observed in macrophages. As the finding presented in the former Supplementary Fig. 9 were essentially also presented as part of the new Fig. 5, we eliminated the former Supplementary Fig. 9 from the manuscript. We would like to note that our new experiments, in which we counted many cells as advised by the reviewer, revealed that complementation of TssG1 from a plasmid in the tssG1 deletion strain did not rescue the phenotype, unlike our previous result that was presented in Supplementary Fig. 9 (see Fig. 5C). We discuss possible explanations for this in the text (lines 250-253).

Minor comments:

The manuscript contains a number of extensive lists that refer to the findings of genome analyses and secretome analyses, respectively. These lists render the manuscript quite difficult to read.

Unfortunately, we do not see a way around this as Vpr evidently possesses a great number of T6SS components and effectors. We hope the visual aids (Fig. 1 and the new Fig. 4) will make it easier for the readers to follow.

In addition to the table (Table 2), it is recommended to present the putative effectors identified by the proteome analysis as a graphic scheme in the main text, highlighting the protein domains of all putative effectors of T6SS1 including the MIX class.

As suggested, we added a scheme of the putative T6SS1 secretome (Fig. 4).

Line 223: „Borkholderia" should read Burkholderia

Corrected.

Effector classes are designated as "clans" throughout the manuscript. Might be better to use the term class?

The designation “clan” is only used when referring to members of the MIX-effector class. We have previously identified this new class of polymorphic T6SS effectors and demonstrated that its members can be divided into 5 distinct clans (named MIX I - MIX V) based on conserved motifs in their MIX sequences (Salomon et al., PNAS, 2014). Therefore, we believe the use of the term “clan” is appropriate.

Page 14: expression of effector in Hela cells and yeast cells: detecting expression of the effector is missing. Please add data in results text.

We added a referral to these data in the text (lines 222-224). We also added a missing panel (Fig EV1D) showing expression of the mutant form of Vpr01570 (Vpr01570^{CS}) in yeast. We note that expression of the wild-type version of Vpr01570 in yeast was not detected, as is sometimes the case for toxins whose expression in yeast is deleterious (Salomon et al., Microbiology, 2013).

A number of text spaces are missing in front of reference brackets. Should be inserted.

Amended

2nd Editorial Decision

07 August 2017

Thank you for the submission of your revised manuscript to our editorial offices. We have now received the reports from the referees that were asked to re-evaluate your study (you will find enclosed below). As you will see, all three referees now support the publication of your manuscript in EMBO reports. Before we can proceed with formal acceptance, I have the following editorial requests that need to be addressed in a final revised version:

You can either publish the study as a short report or as a full article. I suggest the short reports format. For short reports, the revised manuscript should not exceed 35,000 characters (including spaces and references), and should have not more than 5 main plus 5 expanded view figures. The results and discussion section must further be combined, which will help to shorten the manuscript text by eliminating some redundancy that is inevitable when discussing the same experiments twice. Please note that the materials and methods section must remain in the main manuscript file. In contrast, for a normal article there are no length limitations, but the results and discussion section must be separate and the entire materials and methods included in the main manuscript file.

I would suggest a slightly different title (please comment and/or change): Type VI secretion system MIX-effectors have both anti-bacterial and anti-eukaryotic activity

Please add a paragraph to the methods section describing the statistical tests used.

The Appendix figure names and their callout are missing the "S". Please format them accordingly (Appendix Figure Sx).

Please move Table 1 to come before the legends.

Please add a tab to Dataset EV1 and put the legend there. Then cut it from page 36 of the manuscript text.

As already mentioned, we now strongly encourage the publication of original source data with the aim of making primary data more accessible and transparent to the reader (in particular for the Western blot data). The source data will be published in a separate source data file online along with the accepted manuscript and will be linked to the relevant figure. If you would like to use this opportunity, please submit the source data (for example scans of entire gels or blots, data points of graphs in an excel sheet, additional images, etc.) of your key experiments together with the revised manuscript. Please include size markers for scans of entire gels, label the scans with figure and panel number, and send one PDF file per figure or per figure panel.

In addition I would need from you:

- a short, two-sentence summary of the manuscript
- two to three bullet points highlighting the key findings of your study

I look forward to seeing the final revised version of your manuscript when it is ready. Please let me know if you have questions or comments regarding the revision.

REFEREE REPORTS

Referee #1:

The authors have addressed the previous comments. The results are solid. They identified new putative effectors and focused on one effector that carries a predicted toxin domain. The observed effect on actin rearrangement was only detected when it's directly expressed in eukaryotic cells but not when bacteria were added. Overall, it is a solid paper but still suffers from lacking mechanistic insights regarding how effectors are delivered and how the effectors contribute to pathogenesis. However, it seems from the discussion that the authors will tackle these questions in follow-up studies.

Referee #2:

Thank you for addressing my concerns. The revised manuscript is suitable for publication.

Referee #3:

The authors have considerably improved their manuscript and clarity of presentation after responding to three reviewers' comments. I congratulate them to a fine piece of work.

Minor: Quality of presentation of figures: Almost all microscopy pdfs/single panels contain fine vertical and horizontal bright lines. These should be eliminated for the final version.

2nd Revision - authors' response

14 August 2017

We are pleased to submit a final revised version of our manuscript EMBOR-2017-44226V2 in which we include the requested editorial modifications:

- The manuscript has been reformatted to fit the requirements of a Scientific Report (main text with joined Results and Discussion section is now 23,774 characters).
- The title was changed to “**Type VI secretion system MIX-effectors carry both anti-bacterial and anti-eukaryotic activities**”.
- A paragraph describing the statistical analyses has been added to the Methods section.

- The Appendix figure names were amended.
- Table 1 was moved and is now before the figure legends in the main text file.
- A title tab was added to Dataset EV1 and the legend was removed from the main text file.
- Original source data for Western blots were uploaded (Figures 3, EV1, and S1).
- A file containing a short summary and bullet points was uploaded (under file type “Manuscript text”).
- The raw mass spectrometry data that was uploaded to the MassIVE depository has been made public and a link is now provided in the Methods section.
- The Authors checklist was updated.

3rd Editorial Decision - Acceptance

16 August 2017

I am very pleased to accept your manuscript for publication in the next available issue of EMBO reports. Thank you for your contribution to our journal.

YOU MUST COMPLETE ALL CELLS WITH A PINK BACKGROUND ↓
PLEASE NOTE THAT THIS CHECKLIST WILL BE PUBLISHED ALONGSIDE YOUR PAPER

Corresponding Author Name: Dor Salomon
Journal Submitted to: EMBO Reports
Manuscript Number: EMBOR-2017-44226-T+